# Accurate Prediction of Sensory Attributes of Cheese Using Near-Infrared Spectroscopy Based on Artificial Neural Network

**DOI:** 10.3390/s20123566

**Published:** 2020-06-24

**Authors:** Belén Curto, Vidal Moreno, Juan Alberto García-Esteban, Francisco Javier Blanco, Inmaculada González, Ana Vivar, Isabel Revilla

**Affiliations:** 1Department Computer Science and Automation, University of Salamanca, 37008 Salamanca, Spain; bcurto@usal.es (B.C.); jage@usal.es (J.A.G.-E.); fjblanco@usal.es (F.J.B.); 2Department of Analytical Chemistry, Nutrition and Bromatology, University of Salamanca, 37008 Salamanca, Spain; inmaglez@usal.es; 3Department Construction and Agronomy, University of Salamanca, 37008 Salamanca, Spain; avivar@usal.es (A.V.); irevilla@usal.es (I.R.)

**Keywords:** ANN, food sensory prediction, food quality estimation

## Abstract

The acceptance of a food product by the consumer depends, as the most important factor, on its sensory properties. Therefore, it is clear that the food industry needs to know the perceptions of sensory attributes to know the acceptability of a product. There exist procedures that systematically allows measurement of these property perceptions that are performed by professional panels. However, systematic evaluations of attributes by these tasting panels, which avoid the subjective character for an individual taster, have a high economic, temporal and organizational cost. The process is only applied in a sampled way so that its result cannot be used on a sound and complete quality system. In this paper, we present a method that allows making use of a non-destructive measurement of physical–chemical properties of the target product to obtain an estimation of the sensory description given by QDA-based procedure. More concisely, we propose that through Artificial Neural Networks (ANNs), we will obtain a reliable prediction that will relate the near-infrared (NIR) spectrum of a complete set of cheese samples with a complete image of the sensory attributes that describe taste, texture, aspect, smell and other relevant sensations.

## 1. Introduction

With the development of modern society and economy, the food industry needs to guarantee and monitor its product quality to achieve successful commercialization. One way to evaluate the quality and acceptability of a product is the development of a sensory analysis carried out by a panel of judges who are trained using techniques such as QDA (Quantitative Descriptive Analysis), in such a way that the sensory characteristics of the food will be measured. The company, and more concisely its quality manager, will set a specific sensory profiling and the panel will assess the sensory attributes of a food, such as aspect, smell, taste, texture and so on.

Only through the senses, panel members can faithfully determine the sensory attributes of a food (sensory measurement). To guarantee that production process works properly, it would be necessary to regularly perform these sensory measurements inside a powerful quality control policy that would allow detection of the sensory defects when they appear. Unfortunately, these measurements are labor-intensive and would entail high economic, perhaps unaffordable, costs. Also, in many cases, these measurements may become subjective due to the state of the judges (tiredness, sensory fatigue, etc.), and consequently the value of the provided information will be strongly reduced.

An alternative solution would be to obtain trustworthy estimations of these human valuations using physico–chemical data from the food product captured by sensors (instrumental measurements). As [1,2] propose in pioneering fashion, automated quality control procedures can be incorporated into the food processing process. As evident advantages of this approach, the economic, temporal and organizational costs are reduced, so that the devices availability are greater that the human being ones. The reproducibility and objectivity of the results would be obviously increased [3,4].

As an instrumental analytical technique, Near-Infrared Spectroscopy (NIRS) is well known in material sensing. Its major benefit is that it is a non-destructive technique, so if needed (not often) a simple sample preparation must be performed. In fact, it can yield an on-line response for analysis during manufacturing, being rapid, non-invasive, very flexible and robust. NIRS technology applications can be found in medical and biomedical studies, and in forestry, pharmaceutical and petroleum industries [5]. Also, NIRS is used for on/in-line quality monitoring, and safety and authenticity in the agri-food [6], liquid food [7] and meat [8,9,10] sectors, among others.

However, in order to replace human-based sensory panels by artificial systems, it is essential to guarantee accurate predictions. For modeling the relationship between instrumental data and sensory descriptions, classical statistical methods of multivariate analysis, such as MLR (Multiple Linear Regression) and PLSR (Partial Least-Squares Regression) have been considered. However, these methods are based on the linear behavior of variables, so when these variables show non-linear behavior, a situation that normally occurs in data obtained from sensory tests [2,4], the results will not be acceptable. When NIRS technology is used as an instrumental measurement, PCA (Principal Component Analysis) can be applied to reduce the dimensionality of the problem.

To take into consideration these facts, there are approaches widely applied in the food industry for classification and calibration tasks based on NIRS, such as Artificial Neural Networks (ANNs) [6]. This is because they are computer-based modeling techniques that are perfectly suited to discover non-linear trends among variables, especially when these ones are unknown. They can work with signals that contains chaotic and noisy data [11,12]. ANNs allow the design of accurate estimation models that are beyond the reach of classic linear approaches, so they will be considered for our research proposal. ANNs constitute a powerful tool for tasks as predictions due to their powerful learning capabilities that allow the extraction of information for complex datasets. The most commonly used ANN architectures [6] are Multi-Layer Perceptron (MLP), Radial Basis Function, for supervised paradigms and Kohonen Self-Organizing Map for unsupervised paradigm, i.e., when the desired classification is not available.

Within the field of food technology, Neural Networks have been very useful and widely used tools in food safety or food quality analysis [13,14]. More related to the proposed work, ref. [15] uses the chromatographic fingerprint obtained by an electronic nose to predict the physical, chemical, functional and sensory properties of different food products during the different stages of processing and distribution. The synergy of NIRS and Neural Networks has been explored in technology of pork meat [16,17], of wheat flour [18], of wine [19] and other commercial beverages [20]. More specifically, the fatty acid composition of vegetable oils is determined in [21]. The production season and the cows feeding regime in butter production are discriminated in [22]. Also, in a work very closely related to that presented in this paper, an ANN was successfully used in [23] to establish the model for determining the drying time and the percentage of milk mixture (cow, goat and sheep) in cheeses of variable composition from data on fatty acids and NIRS. As a summary, it can be stated that most of the work is aimed at classifying and predicting the composition and processing conditions of different food products.

In this paper, we present a study on the use of ANNs for the estimation of 19 sensory characteristics of samples of controlled-processing cheese. As instrumental measurements we use the NIR spectrum in the range 1100 to 2000 nm. The NIR information was coupled with 19 sensory attributes to predict aspect, taste, smell, texture, and other sensations linked to cheese quality. The training and the prediction ability of the ANN performance is based on all the data provided by a panel of judges trained in the QDA methodology.

## 2. Material and Methods

The global schema of the proposed system to make the prediction of the sensory attributes of a cheese sample can be observed at the Figure 1. The system has as input, the NIRS data of the sample and as output, its sensory properties, described by 19 variables. The following sections will outline the main elements of our proposal, together with the material and methods used. First, the material used will be described: a complete set of samples, which were taken from a wide range of cheeses made from sheep’s, cow’s and goat’s milk in different percentages. The sensory analysis of the samples carried out by a panel of experts in QDA methodology is then described. As an instrumental technique, NIR spectroscopy was applied to each of the cheese samples to extract their spectral information. Next, the two main techniques that we have considered will be presented, i.e., Principal Component Analysis (PCA) to perform a reduction of the dimensionality of the input data, and Artificial Neural Networks (ANNs) to make numerical predictions for each of the sensory properties.

### 2.1. Cheese Samples

Various types of cheese were produced in a controlled manner in the pilot plant of the Food Technology Area at the University of Salamanca. Two production processes were carried out: 112 cheeses in the winter season and 112 in the summer with milk collected directly from farms located in Zamora (Spain). In a prescribed way, cheeses of 16 different formulations, between 0% and 100%, of raw milk from cow, sheep and goat were made [24]. After a classic initial production process, several pieces of each formulation were taken, initially with a diameter of 10 cm and a thickness of 5 cm. The aging process was performed at a pilot plant with controlled climatic conditions (15 ${}^{{}^{\circ}}$C and 70% HR). In this way, a period of seven (7) months of aging (from 0.2 to 6 months) was considered to take into account this characteristic in the sensory analysis procedure. In this work, cheese samples of each of the 16 formulations from winter and summer milks and two (2) aging periods (4 and 6 months) were considered to perform spectral (NIR) and sensory analytical testing, with a total of 64 different samples.

### 2.2. Sensory Analyses

In [25], the sensory task is described in detail and rigorously, including the conditions of the activities performed. In the present work, we only use the complete data provided by [25]. Therefore, only the most significant details needed to understand the data used in our work will be described below.

A panel of 8 judges was trained in the application of the QDA methodology over 18 sessions. During the training, the panelists agreed on the benchmarks, terminology definitions and evaluation techniques in the sensory profile of the cheese. Sensory properties related to aspect, taste, smell, texture and other sensations were considered. Specifically, there were 19 attributes evaluated (Table 1), framed within: aspect (homogeneity, color, holes), taste (salty, rancid, intensity, sour, buttery), smell (rancid, lactic acid, intensity), texture (chewiness, creaminess, fatty feeling, granularity, hardness) and other sensations (hot, remains in throat, retronasal).

To quantify the intensity of each attribute, 8-point scale was used, with ‘0’ corresponding to the absence of the parameters, ‘1’ referring to the minimum intensity and ‘7’ to the maximum intensity for each of the parameters. This type of scale was selected because it had given good results in other previous works, when it was used to evaluate the characteristics of the cheeses [26]. To avoid scaling problems, the tasters’ scores for each characteristic have been normalized inside the interval from −1 to 1, based on the maximum and minimum values of the opinions made by the judges.

### 2.3. NIR Spectroscopy

NIR spectra were acquired by a Foss NIR 5000 with a standard 1.5 m 210/7210 bundle fiber-optic probe (Ref no. R6539-A). The probe employs a remote-reflectance system and has a window of quartz with a 5 cm ×5 cm surface [27]. NIR spectra were recorded by directly applying the fiber-optic probe to slices at least 1 cm thick at room temperature (20–23 ${}^{{}^{\circ}}$C). Therefore, this instrumental measurement did not require any sample preparation.

For each sample, 32 scans were performed, which were averaged to give a spectrum represented as values of log(1/R) (*R* means reflectance) as a function of the wavelength in the range between 1100–2000 nm with a nominal spectral resolution of 2 nm. To minimize sampling error, all samples were analyzed in triplicate. Prior to each recording, the probe window was cleaned to minimize cross-contamination. In NIRS spectra, we obtain the reflectance measured against the wavelength in the near-infrared region, providing chemical-physical data about the food product. These parameters characterize, although not directly, the sensory properties of food as perceived by consumers. As a result, spectral information can be used to determine the sensory attributes of cheese.

### 2.4. Principal Component Analysis

NIR spectroscopy constitutes a technology that generates a huge amount of data about organic materials. PCA is useful to translate the concrete information to a set of combinations, named principal components (PC), that explain the variability of the dataset.

As has been exposed, 64 samples of cheese were used for this work. At previous section, the wavelength interval used for NIR was presented, so for each sample, 450 variables are considered, i.e., log(1/R) for each wavelength. The complete set of NIR data for all 64 samples will be taken into account. Our goal is to reduce the dimensionality of the variables, so the ANN model will be feasible. Instead of inputting 450 variables into the ANN, we tried to find a few representative ones of the spectral information: the PCs that explain the variability of the dataset.

### 2.5. Artificial Neural Networks

The mathematical tool used to predict the sensory characteristics of cheese samples is based on ANNs, which fall into the non-linear modeling methods.

ANNs are made up of process elements (artificial neurons) distributed in layers linked by weighted interconnections. The MLP (Multi-Layer Perceptron network) architecture has been successfully used to make accurate estimates in our work and, as it has been mentioned previously, in multiple works in the field of food technology [13]. It has three layers. External data are entered in the input layer, in our case the information is related to the NIR spectra of the cheese samples, processed by PCA. Once the data are processed by the neurons in this layer, they spread to the middle layer. The neurons in the hidden layer produce intermediate results that are provided to the output layer to calculate the ANN response. For each sensory attribute, an ANN model will be constructed: in the input layer, the PCs data of the cheese sample are entered and in the output layer the prediction of a sensory attribute of the cheese sample is obtained. Therefore, each ANN model will be designed using the JavaNNS (Java Neural Network Simulator) application and the model will be developed within the application as a Multi-Layer Perceptron network (MLP).

The parallel processing of the different neurons allows large volumes of data to be processed. The information processed in each layer is propagated as a wave of data between the three layers, from left to right, through the existing connections between the neurons in each layer. Each pair of neurons in different layers is connected by a weight that indicates the degree of interaction between the neurons. The output of a neuron depends on an excitation function (linear or not) of the neuron and is the sum of the input values, weighted by the connection weights.

The ANN can adapt to the environment, through examples, by adjusting the weights that ponder the connections of the neurons. One of the learning paradigms of the network is supervised learning, where the desired behavior is taught through training examples, in our case the sensory score dataset provided by trained panelists as output, and the spectral dataset as input. The ANN processes the input data that belongs to the training dataset, and compares the outputs produced with those expected. The network iteratively adapts the weights (initially they have a random value) of its connections to minimize a function of the error between the result of the ANN and the expected result of the training dataset. At the training stage, a backpropagation algorithm will be used where the calculated error is propagated backward through the network layers.

A training data instance will be a $\left(x_{i},y_{i}\right)$ pair, where $x_{i}$ will be related, using PCA, with the NIR measurement for a cheese sample and $y_{i}$ will be sensory score given for a human tester. Therefore, the training dataset will be constituted by several pairs $\left(x_{i},y_{i}\right)$.

Finally, at the validation stage, an attempt will be made to assess the capacity of the trained network to generalize. For this purpose, a set of samples is used, a validation dataset, which is representative and not used during the training stage. In this way, we try to avoid over-training the network, so that it will have the capacity for generalization. A five-fold training procedure will be applied, where a split is made with a selection of 20% instances for validation (randomly selected) and 80% for training, with five (5) training experiments performed. In this way, the size of the training set, in addition to the network architecture, has influence on its generalization capacity, so it must have a sufficient number of samples and it will be related to the number of neurons in the hidden layer.

### 2.6. Metrics to Evaluate the ANN Model

Once the ANN has been trained, an prediction model will be available, making it necessary to evaluate its performance. In this study, a set of indexes and plots will be computed to evaluate the neural model performance for an unknown set of *N* samples (validation dataset):The root mean squared error of prediction (RMSEP)
$$RMSEP=\sqrt{\frac{\sum_{i=1}^{N}{\left(y_{i}-\hat{y_{i}}\right)}^{2}}{N}}$$
being $y_{i}$, $\hat{y_{i}}$ the judges’ real score and the prediction (artificial score) of an attribute of the sample *i*-th, respectively.The squared correlation coefficient $R^{2}$ defined as
$$R^{2}=1-\frac{SSE}{SST}$$
where SSE is the Sum of Squared Error between the model predictions ($\hat{y_{i}}$) and the target values $y_{i}$, i.e., the mean of the judges’ individual scores for each sample
$$SSE=\sum_{i=1}^{N}{\left(\hat{y_{i}}-y_{i}\right)}^{2}$$
and SST is the Total Sum of Squares defined as the sum of the squares of the difference of the judges’ scores and its mean $\bar{y_{i}}$
$$SST=\sum_{i=1}^{N}{\left(y_{i}-\bar{y_{i}}\right)}^{2}$$Consequently, a perfect fit is given when $R^{2}=1$ and there is no fit when $R^{2}=0$.Model residuals plot, i.e., residual defined as the difference between predicted and measured value ($\epsilon_{i}=\hat{y_{i}}-y_{i}$).Scatter plot of predicted values and judges’ individual score mean for each sample.

## 3. Results

In the following, the results that are concerned in our work will be exposed. First, the data that will be used to train and to assess the ANN from the tasting panel will be presented. Then, NIR spectra obtained from cheese samples are presented. PCA will be applied to this dataset, i.e., the spectra, to obtain a set of principal components that will be used as input to the ANN. Finally, the analysis of ANN performance over the validation dataset will be shown.

### 3.1. Measurements of Taster Evaluation Results

In the sensory evaluation phase, the panelists graded the cheeses according to the list of descriptions defined during the panel training. In particular, the 10-member panel met in 4 tasting sessions to evaluate the cheeses. These 4 sessions correspond to cheeses with 4 and 6 months of maturation and made with winter and summer milk. In each session, cheeses of the 16 formulations were tasted and the 19 attributes described previously were evaluated by the judges. However, scores of 2 members were discarded as strongly disagreeing. In total, for each of the attributes, a score dataset of 512 items is available.

Table 1 includes the 19 sensory attributes evaluated by the panelists and the main statistical results (mean, average, minimum and maximum) relating to the tasting activity. These statistical results have been calculated from a set of 512 individual scores, corresponding to the sensory evaluation of cheese of the 16 different formulations in 4 tasting sessions by the 8 judges.

### 3.2. NIRS Measurements

In each tasting session, the spectrum for the complete set of cheeses was recorded and in Figure 2 the results can be observed. The trends of spectra were quite similar. The entire wavelength range of 1100–2000 nm was applied to develop the PCA-ANN prediction models. The spectra curves have an increasing global trend with 4 strong local peaks at wavelengths around 1210, 1450, 1730 and 1930 nm and 3 local valleys around wavelengths of 1280, 1650 and 1850 nm. We consider that no pre-processing of the spectrum data is necessary to avoid any loss of information. Therefore, the complete and original dataset will be used with Principal Component Analysis.

### 3.3. Principal Component Analysis

Our goal was a dimensional reduction for the spectral information so that a significant reduction for the number of input variables is reached. As the NIR spectrum extends from 1100 to 2000 nm with a spectral resolution of 2 nm, it produces 451 values. Applying the PCA technique, three (3) principal components that can explain the 99.98% of spectral variability are obtained. PC1, PC2 and PC3 explain most of the variability of the spectrum data: 86%, 12% and 1% respectively. Loadings of each component are constituted by 450 values, i.e., its relationship with the considered 450 variables (each one related with the reflectance at a given wavelength), so loadings of component would be seen as function of the wavelength.

Figure 3 (loadings plot) shows the weights for each the wavelength (variable) when the main components are calculated: the three PCs summarizing the information retained in the whole spectral dataset. This situation can be also observed graphically in Figure 4 (scores plot), where the main variations appear along the first component axis, followed by variations in the second component. Finally, at the third component minimal variations can be observed.

### 3.4. ANN for the Sensory Prediction

At Figure 5, the topology of the ANN model can be observed, with three (3) layers connected by feed-forward links. This model is constructed in the following way:An input layer of 3 processing elements (PE or neurons) that uses the linear function as an activation way. The inputs will be composed by the three (3) principal components that are obtained using PCA. In Section 3.3 the description of these components has been presented.A hidden layer constituted by 5 PEs. According to Kolmogorov’s theorem, a three layer ANN with 2n+1 PE (where *n* is the number of inputs) in the hidden layer is enough to map any function, so we would need 7 elements in this unique hidden layer. However, in our case, the size was reduced to 5 neurons to avoid an overfitting risk. We have used the hyperbolic tangent function as an activation function. As it is known, it is responsible that non-linear behaviors can be learnt.An output layer with one (1) PE that provides the artificial score of the modeled sensory characteristic. It has a linear behavior.

Therefore, nineteen (19) Artificial Neural Networks have been trained: one for each sensory characteristic. The training process described below has been developed independently for each ANN.

Each network instance uses a new initializing process for the values of the link weights. As a metric of the training parameter performance, the mean squared error (MSE) on the training dataset has been used, so the training algorithm is stopped as soon as this value no longer decreases. Also, it must be avoided that the MSE can increase, i.e., due to an overfitting scenario in the ANN learning process. In the backpropagation with momentum algorithm used, values μ=0.05 for the momentum and a learning rate δ=0.00002 have been considered. As for each network the process has been developed independently.

At the end of this phase, a NIRS-based prediction model is available for each sensory attribute *p*, where for the sensory attribute value $y_{i}^{p}$ of the sample *i*-th, a prediction ${\hat{y_{i}}}^{p}$ is obtained with an error $\epsilon_{i}^{p}$
(1)$${\hat{y_{i}}}^{p}=f^{p}\left(x_{i}\right)=y_{i}^{p}+\epsilon_{i}^{p}$$
where $f^{p}$ will be considered to be a function for each attribute *p*, being p∈1,…,19; $x_{i}$ is related to spectral data of the sample *i*-th and to the three (3) PCA components that can be obtained from the NIR spectra for the complete set of cheese samples. The prediction model that will be considered the most accurate will be the model that minimizes a function of the error ($\epsilon_{i}^{p}$). In this way, ref. [28] defines “accuracy” as closeness to the panelist judgement (reference data).

For the training and validation of each of the 19 ANNs, a 512-item score dataset has been available: 64 samples that are tasted by 8 judges. Therefore, for each parameter, we have 512 pair $\left(x_{i},y_{i}\right)$, with (8) different values $y_{i}$ for each sample. This set has been split into 80% of the items for the training phase and 20% for the validation phase of each ANN, as explained previously. This number of items in the training dataset and in the validation dataset has been sufficient to guarantee a reliable ANN model, as we will check when evaluating the performance of the resulting ANNs.

### 3.5. Evaluating the Performance of Sensory Attribute Predictions

In our work, a set of indexes and plots, as explained in the “Materials and Method” section, has been computed to assess the performance of each neural model in the validation dataset (unknown samples).

As it can be observed in Figure 6, RMSEP can be found between a value (0.34) for taste intensity, fatty feeling and chewiness and a value (0.59) for granularity, except for holes. These low values that are based on the error function, lead to the conclusion that the ANN model is very accurate, not so much for holes.

The Figure 7 shows that for the squared correlation coefficient $R^{2}$ obtained for each one of the 19 trained networks, with exceptions, the obtained values are between 0.73 and 0.87. As a clear conclusion, it supposes a high level of adjustment. As can be seen, the networks that achieve a better fit correspond to those that predict the intensity of taste, chewiness and fatty feeling (texture) with values of 0.87, 0.86 and 0.87. In the opposite, the lower values of $R^{2}$ are obtained in the attributes: rancid (smell), salty (taste) and remains in throat with values 0.73, 0.73 and 0.74, except for buttery and holes.

To illustrate the predictive capacity of each ANN, the artificial scores provided for each parameter are presented (Figure 8—scatter plot) on the same graph with the mean judges’ scores for the prediction dataset. It can be observed that ANN predictions are mostly inside an acceptable interval around the expected value for the set of taster evaluations, pointing out that the expected value is defined by the mean per sample of the individual judgements. This interval is marked within each figure by red color lines, defined by the deviation in judges’ scores for each attribute, with the data collected in Table 1. Therefore, ANN prediction behavior is similar than taster.

As another way to review the assessment of the prediction capacity of the ANNs developed in this paper, a set of plots of the obtained residuals is presented, i.e., the difference between the total mean of the assessments made by the judges and the value estimated by the ANN for each sample of the validation set. At Figure 9, only the results for the attributes with the highest and lowest values of $R^{2}$ are shown. The values of residuals for the all characteristics are mostly within the range (−0.5,0.5) around the estimation value, with mean values of residuals between 0.24 (chewiness) and 0.39 (salty), which clearly means that the estimation value is correctly adjusted to the human tasters assessment. It must be remembered that these valuations are quantified with integer values between 1 and 7 (Table 1).

## 4. Discussion

The obtained results can be considered sufficiently successful. There are few studies dedicated to the application of ANNs to model the relationship between sensory analysis and instrumental measurements. As one of the preliminary works, in [29] ANNs were applied to the determination of mathematical relationships linking human sensory judgements to physical measurements of external color for tomato and peach. The involved parameters, concerning the perception of fruit color, are directly related to the instrumental measures, instead of our case where the experimental measurement is not directly related to any of the sensory parameters, because there is no direct property measurement (holes for example). Other related work, ref. [11], tries to rationalize the use of concentrates in fruit drinks production, so they constructed an ANN model to make predictions about the taste intensity of blackcurrant concentrates using gas chromatography as an instrumental measurements of these flavor component but, evidently, these ones are directly related with the perception (flavor intensity). Anyway, as in our work, they show that ANN performance is better that PLS or PCR methods. Another application of ANN that is related with sensory properties can be found at [30] that makes predictions about the formulating process for food products (such as beverages) based on their chemical composition. The correlation between sensory and instrumental data was studied using ANNs but combined with classical methods of data analysis based on linear models. It shows the capabilities of ANN, especially when the number of involved variables grows.

More directly related with our proposal, in [31] MLP was applied to evaluate the sensory texture properties of cooked sausages, but, first the dataset size is reduced, and the instrumental measurements (mechanical laboratory tests) are directly related with the sensory properties. We must indicate that in our case the relationship between sensory parameters and NIRS is not so direct. Also, in the same way, ref. [4] makes use of MLP to make predictions of the texture attributes of light cheesecurds perceived by trained judges based on instrumental texture measurements. At [12] a relationship between chemical parameters, such FFA (Free Fatty Acids) or PV (Peroxide Values), with sensory panel results to distinguish the olive oil quality (extra virgin or no extra virgin) is proposed. It is a classification problem, in the same way our group obtained good results with our same cheeses in a previous work [23]. More recently [32] classifies strawberry purees using spectral information and Deep Neural Networks and [33] also performs classifications tasks. Thus, many studies focus only on classifying food according to quality criteria or on determining only one attribute by means of a non-online instrumental measure, but not on the determination, using NIRS, of all the attributes with numerical values that the panelists have evaluated.

The spectral data were incorporated into the ANNs by means of Principal Component Analysis. With the first three components, 99.98% of the spectral variability can be explained: 86%, 12% and 1%, respectively. From a comparative point of view, the weight of the third component PC3 may be negligible compared to the weight of PC1 and PC2. However, we have also included the third component as an input in the ANN, since we think that in a non-linear behavior this small weight can be significant, and the third component can incorporate useful information.

For the preparation of the ANN, we have had 19 data sets with 512 items in each: the scores of 19 sensory parameters given by 8 judges for 64 different cheese samples. Therefore, we have used 512 pairs (spectral data input, attribute score output) with (8) different values of attribute score output for each sample. In our preliminary experiments for the training of each ANN, we only considered the mean of the tasters’ scores for each sample and the results were very poor. Later, when we considered the entire dataset (512 items), with the individual scores of each judge, the training of each ANN was possible, and the results improved significantly. The individual data provided by taster (with the individual noise) played a positive role in the realized training experiment, no resampling activity is needed, nor repeating the samples to adjust the probability distribution of the data.

In any case, we must realize that two parameters (holes and buttery) have acceptable behavior, but it is worse than the rest. It must be considered that *holes* present largest deviation than the rest and *buttery* has reduced span in the taster reactions (Table 1). Anyway, research work must be performed in the future to find the activities to enhance the prediction results for them.

Considering the same task and the same dataset, we then make a comparison with other techniques to make the predictions. It can be observed that our results, obtained with ANNs on unknown samples, are better than those obtained in [25]. Our proposal offers mean values for residuals between 0.24 (chewiness) and 0.39 (salty) and [25] presents values between 0.9 for holes and 0.4 for creaminess, chewiness and retronasal that in fact constituted good results.

We think that the reason is that MPLS (Modified Partial Least-Squares) that they used is a procedure based on linear models, and it can be posed that the relationship between sensory parameters and NIRS has a non-linear behavior. Our future work will be related to the considerations of other varieties of cheeses to improve the usability of the approach. Also, the use of more complete information as hyperspectral as [34] can be very useful in this kind of estimation. It could be interesting to include the aging feature as a differentiation in the prediction. We think that it could be quite useful, and it will be considered in future works.

## 5. Conclusions

The functional relationship of the numerical values of sensory attributes for food products with the instrumental measurements obtained with current techniques, such as NIR, can be posed as modeling task of a highly complex relationship. This is due to the fact that the mechanisms that relate these datasets are multiple and each of them will be complicated (or unknown), so the model will probably be not linear. In this sense, ANNs are the ideal modeling tool because their bases use this fact (use of non-linear functions) as a key aspect of its working way.

In this paper, a serious experimental work on the determination of sensory attributes has been presented and it is supported by an acceptable volume of evidence on a well-defined population. These data and the realization of experimental NIR scans on the food (cheese) samples will constitute the data that will be used for the learning task. As a final tool, the obtained system based on PCA-ANN allows reliable predictions of sensory attributes of the samples, with better results than MPLS on the same experimental data sets. As a result, a very useful tool for quality control with a very low computation cost is obtained. In addition, it must be considered that the proposed procedure can be improved, including other varieties of cheese as well as the aging feature as a prediction, at future works.

## Figures and Tables

**Figure 1 sensors-20-03566-f001:**
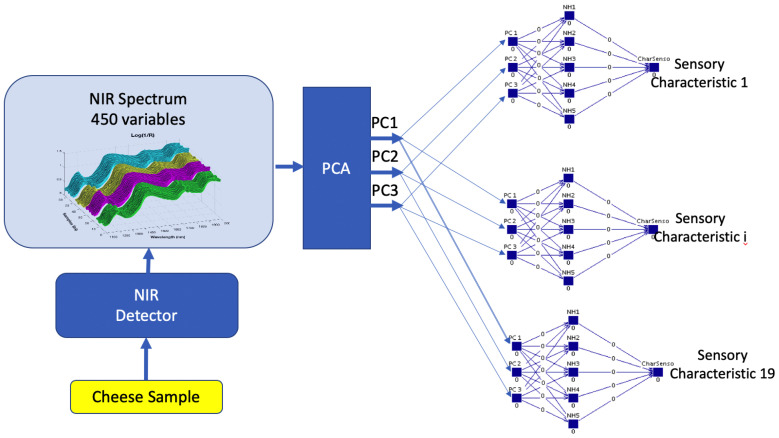
Global scheme of the prediction of 19 sensory parameters for each cheese sample. PC1, PC2 and PC3 denote the three principal components of Principal Components Analysis (PCA) on NIR spectra. Sensory characteristic *i* refers to each of the 19 sensory parameters of the cheese to be estimated.

**Figure 2 sensors-20-03566-f002:**
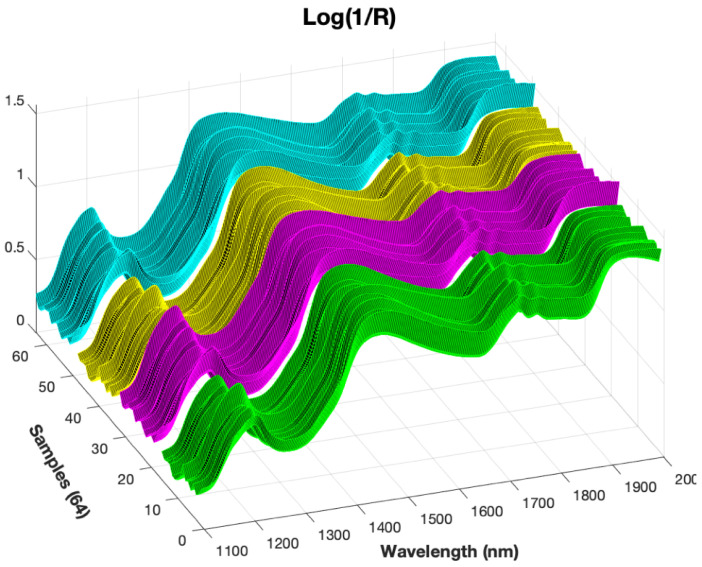
NIR measurements (spectra) for the whole sample set in the wavelength range 1100–2000 nm. Four colors are used to distinguish the spectra according to the aging and seasons of the cheese (4 months/Winter-Green, 6 months/Winter-Magenta, 4 months/Summer-Yellow, 6 months/Summer-Cyan).

**Figure 3 sensors-20-03566-f003:**
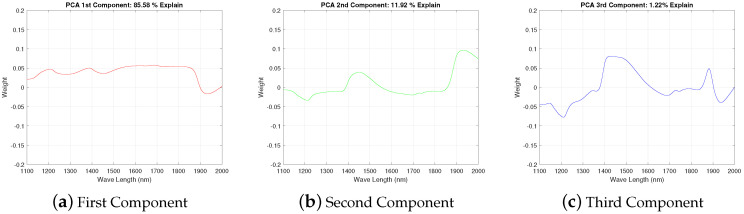
Loadings plot—Principal Components obtained by PCA from NIRS. The weights assigned to each one describes the relative influence for each wavelength.

**Figure 4 sensors-20-03566-f004:**
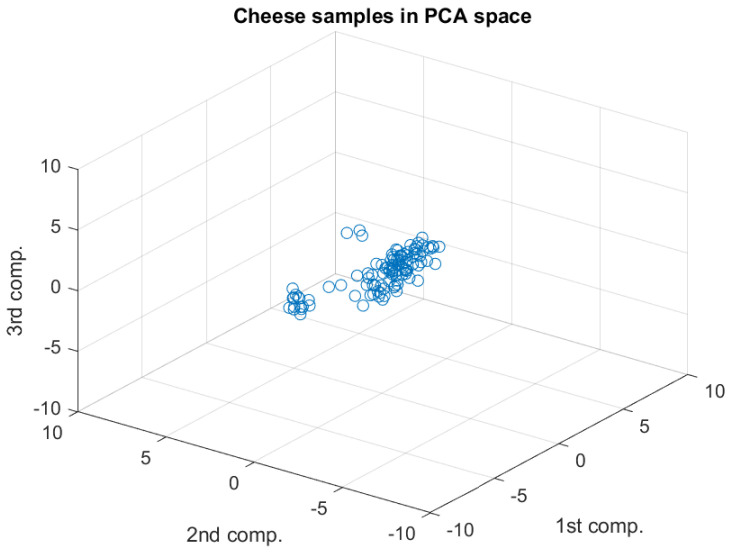
Scores plot—cheese spectral dataset represented in the Principal Components space. It can be observed that the main variability is found along the first PC. In the third direction the variation is quite small.

**Figure 5 sensors-20-03566-f005:**
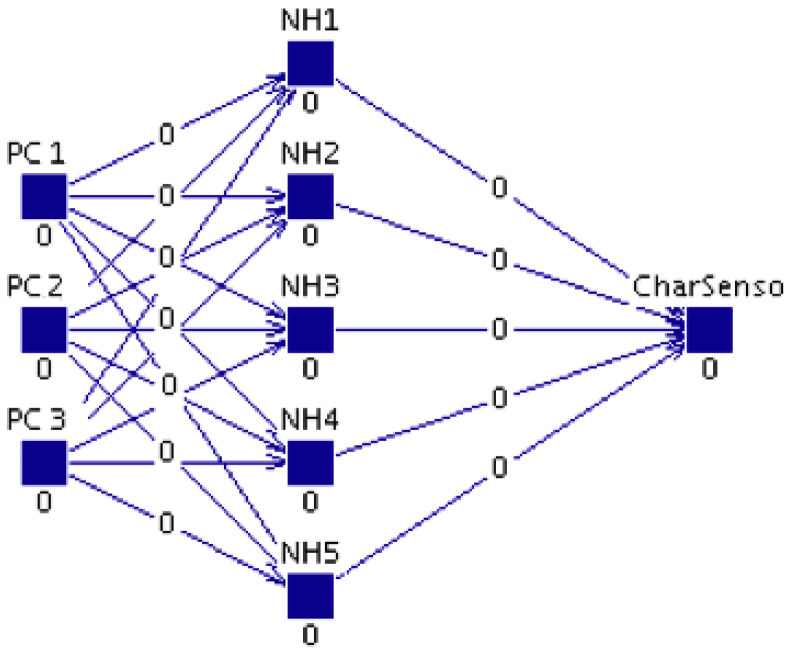
Schematic representation of neural model for the prediction.

**Figure 6 sensors-20-03566-f006:**
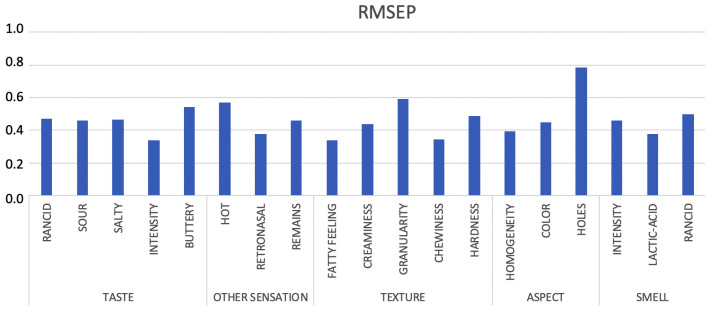
The root mean squared error of prediction for the whole set of parameters for unknown samples.

**Figure 7 sensors-20-03566-f007:**
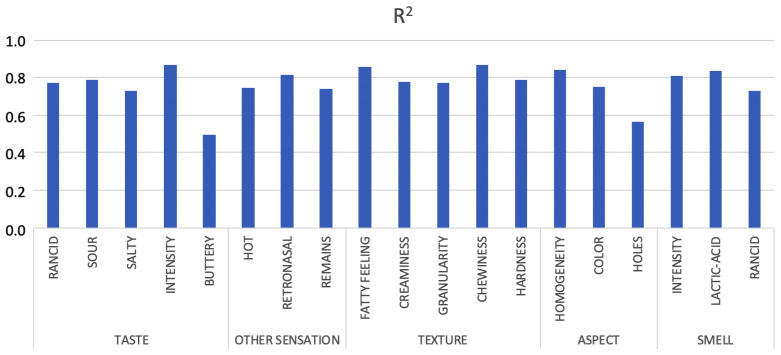
Squared correlation coefficient of prediction for the whole set of parameters for unknown samples.

**Figure 8 sensors-20-03566-f008:**
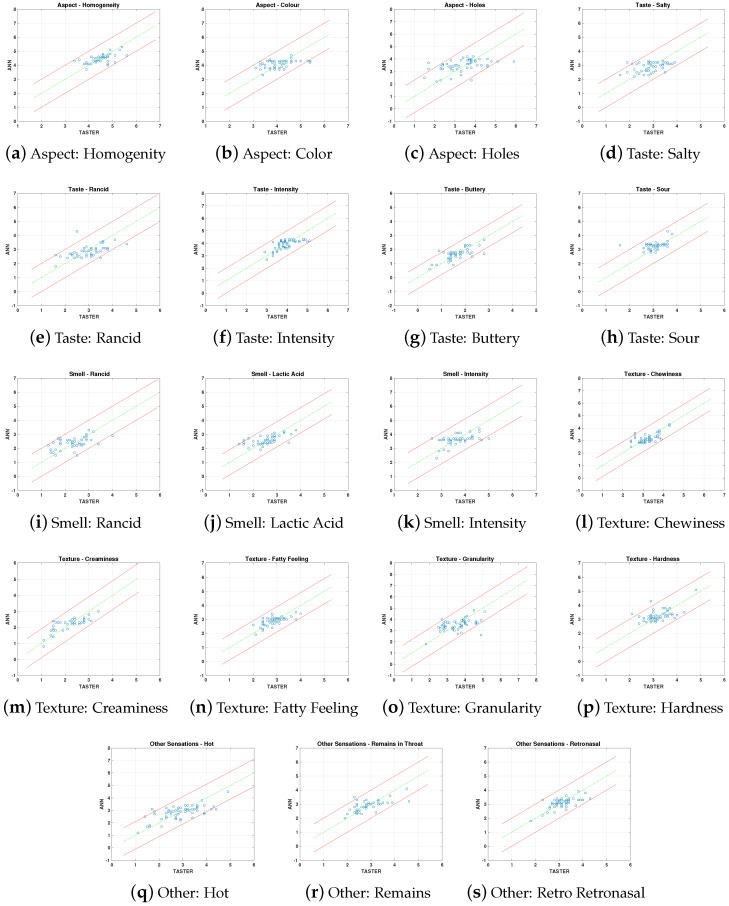
Scatter plot for PCA-ANN prediction results vs tasters’ scores for all sensory attributes for the prediction dataset. The performance over the attribute span is observed for the whole set of attributes in the prediction stage. The green line is the expected score and the red lines mark the interval defined with the deviation of the tasters’ scores for the total set of samples.

**Figure 9 sensors-20-03566-f009:**
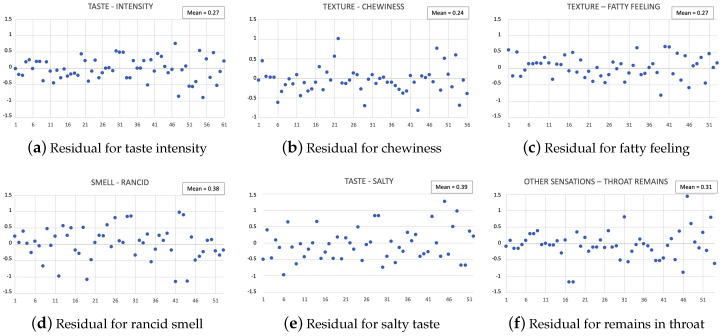
Residual for the predictions for different atributes.

**Table 1 sensors-20-03566-t001:** Sensory parameters applied by the taster panel and their main statistical descriptions for the total sample dataset considering judges’ individual scores.

Property	Mean	Deviation	Minimum	Maximum
Aspect-Homogeneity	4.5	1	1.5	7
Aspect-Color	4.1	1	1.5	7
Aspect-Holes	3.4	1.3	1	7
Taste-Salty	2.9	1	1	6
Taste-Rancid	2.8	1	1	6
Taste-Intensity	4	1	1	7
Taste-Buttery	1.7	0.8	0	4
Taste-Sour	3.2	1	1	6
Smell-LacticAcid	2.6	0.9	0.5	6
Smell-Rancid	2.4	1	1	6
Smell-Intensity	3.7	1.1	1	6
Texture-Chewiness	3.2	0.9	1	6
Texture-Creaminess	2.2	0.9	0	4.5
Texture-Fatty Feeling	2.9	0.9	1	5.5
Texture-Granularity	3.5	1.2	1	7
Texture-Hardness	3.2	1	1	6.5
Other-Hot	2.9	1.1	1	7
Other-Remains	2.9	1	1	7
Other-RetroNasal	3.1	1	1	6

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
