# Peer review of "Accurate Prediction of Sensory Attributes of Cheese Using Near-Infrared Spectroscopy Based on Artificial Neural Network"

_sensors, 2020, doi:10.3390/s20123566_

Round 1
Reviewer 1 Report
The paper bases on a innovative but difficult task: predict sensory properties, as determined by sensory analysis, by near infrared spectroscopy. The authors presented cheese as case study.
Even though the novelty and originality of the study is relevant, the paper lack of the proper construction. In different cases, as reported in detail below, part of the text should be moved from results to M&M. In any case both sections lack of relevant information as details in the comments below. Moreover, the discussion and results are quite poor in addressing the key point of the research.
The poor English form and the typo made difficult the reading in different paragraphs. Therefore, a major revision of the manuscript is advice.
Introduction
Few consideration about the work cited:
- Instead of ref [3], consider to cite a more recent publication of the same research group. This work consider different animal product analysis (meat, poultry and fish) and led to an ad hoc e-nose system.
Grassi, S., Benedetti, S., Opizzio, M., di Nardo, E., & Buratti, S. (2019). Meat and Fish Freshness Assessment by a Portable and Simplified Electronic Nose System (Mastersense). Sensors, 19(14), 3225.
- I would suggest to substitute the reference [8] with a review in the meat science field, instead of citing a research paper. No interest in one specific, I just suggest few of them, but it is up to the authors the selection.
Chapman, J., Elbourne, A., Truong, V. K., & Cozzolino, D. (2020). Shining light into meat–a review on the recent advances in in vivo and carcass applications of near infrared spectroscopy. International Journal of Food Science & Technology, 55(3), 935-941.
Kademi, H. I., Ulusoy, B. H., & Hecer, C. (2019). Applications of miniaturized and portable near infrared spectroscopy (NIRS) for inspection and control of meat and meat products. Food Reviews International, 35(3), 201-220.
Pasquini, C. (2018). Near infrared spectroscopy: A mature analytical technique with new perspectives–A review. Analytica Chimica Acta, 1026, 8-36.
Weeranantanaphan, J., Downey, G., Allen, P., & Sun, D. W. (2011). A review of near infrared spectroscopy in muscle food analysis: 2005–2010. Journal of Near Infrared Spectroscopy, 19(2), 61-104.
Prieto, N., Roehe, R., Lavín, P., Batten, G., & Andrés, S. (2009). Application of near infrared reflectance spectroscopy to predict meat and meat products quality: A review. Meat science, 83(2), 175-186.
- Please consider to reduce the examples reported for the use of FT-NIR spectroscopy combined with a Back Propagation ANN for the simultaneous analysis of multi-components in food. In detail, consider to reduce the paragraph from L67-L76
Please consider to reduce the presentation of the aim in the introduction. An attempt is reported below:
L84-90: In this paper, we present a study on the use of ANNs for the estimation of 19 sensory characteristics of samples of controlled-processing cheese. As instrumental measurements we use the NIR spectrum in the range 1100 to 2000 nm. The NIR information was coupled with 19 sensory attributes to predict aspect, taste, smell, texture, and other sensations linked to cheese quality. As a key element of learning process, the training and the prediction ability of the ANN performance is based on all the data provided by a panel of judges trained in the QDA methodology.
Material and Methods
More details should be provided about the number and weight of cheeses produced. How many cheeses were produced for each formulation? Which was the size of each cheese?
More detail about the amount of sample testes by the panel and by NIR analysis should be provided. How many slices were analysed by NIR, in which point? The NIR signal is highly influence by water distribution and more other changes occurring in the lipid fraction, but this changes are highly influenced by the cheese (sample) geometry, thus further details should be provided. A Figure could better explain the procedure. Also the ANN description is not exhaustive. Figure 2, 3 and 4 seems not to be relevant in the material and methods section, consider to remove them.
Moreover, specific comments are reported in detail below.
L108: Could be an idea to call the 16 different cheeses, 16 different formulations
”16 different cheeses were made with percentages between 0% and 100% of raw milk from [...].”
L110: To what referred the authors with “elaboration process”, production process?
L113: Consider to substitute “laboratory (NIR)” with “spectral (NIR)”
L114-115: Please considered to remove: “Therefore, we have worked with cheeses of 16 different compositions, with milk from two different seasons and different drying times.”
L125-127: Here the authors refereed to pictures of the cheeses to evaluate the attributes of appearance. Is it possible to provide more details? Were the pictures produced on the considered formulations? Are the pictures reference materials? In case they were shot by the authors, details should be provided about the place (black box), illuminations and other environmental and resolution conditions.
L130-138: Please provide more details about spectral resolution and/or integration time used to obtain the spectral information. It would be an idea to move here the information reported at line 142 (from 1100 nm to 2000 nm with a separation of 2 nm). Indeed, these information are linked to the NIR analysis and not to the PCA elaboration.
L141: Please modify here and in Figure 1 the expression “set of values”, this is a set of variables
L140-144: please indicate here the number of samples included in the PCA. An idea could be to write that the data matrix analyzed by PCA was composed by ‘n’ number of samples and ‘m’ variables, that is your 450 variables. Were the data transformed prior to multivariate data analysis? Spectral data, sensory data were pre-treated? Mean centering, Autoscaling which transformation was used? Correction of the spectral signal, i.e. smoothing, baseline correction, MSC, SNV?
Information about the software used should be provided here and in the ANN section.
Artificial Neural Networks section. This section is not well structured, different details are reported but relevant information are missing or not well described. For instance, from Figure 1 it is clear that PC1, PC2 and PC3 were used, but here no information is provided. No detail about the number of samples used in calibration and prediction is provided and so on. The method description should be highly improved.
Results
Measurements of Taster Evaluation Results: Could you describe if the scale was anchored or not.
L188: Could you explain the meaning of this sentence: “Then, NIRS will be shown with the following PCA”
L205-209. This sentence should be reported in Material and Methods
Figure 2 should be moved here, as here it is cited and commented. Indeed, spectral signal should be described and associated to chemical information. Spectral should be coloured according to type of cheese and/or maturation time. There is no need of the colouring solution proposed by the authors.
L219: Please move here Figure 3. The sentence “Figure 3 shows these three components over the wavelength.” Is not clear, these are the three PC summarising the information retained in the whole spectral dataset.
L220-221: They explain most of the variationship at the spectrum: 86 % 12 % and 1 % respectively. What does it means variationship? Variation? Variability? By PCA algorithm the PC1 explains the higher amount of variability and each new component explains less and independent variability (as they are orthogonal). This is by principle, it should not be explain in this way here. Figure 3 and 4 should be moved here and described in detail as loadings plot (Figure 3) and scores plot (figure 4). A good explanation of both graphs is missing. Describe the shape of the spectra, the peaks, the baseline changes for the loadings plot. Actually PC1 seems to describe a baseline variation; the baseline was corrected somehow? Otherwise the information described is more linked to the roughness of the cheese slices than to real changes in the cheeses. In Figure 4 samples appears quite low in number, how many samples were analysed? For a reliable ANN model you should guarantee an high number of samples in calibration and a worth number in prediction. Samples in the scores plot should be coloured according to a “category”. As mentioned for Figure 2, they could be coloured by type of cheese, maturation time, one or few sensory attribute(s), for example.
L225-227: This sentence should be moved in M&M.
L249-L251: This sentence is relevant for M&M section to describe the data transformation
L259: Finally here it is reported that the prediction set was composed by 20% of the data. This information should be reported in M&M with the number of data collected, as mentioned in the previous comments.
Figure 8. No matter the figure of merits reported (RMSEP and coefficient of determination), it looks that the prediction vs actual values lines (predictors) are quite poor for all the 19 parameters, could the authors explain this?
L274-276 How were acceptable ranges defined?
L277-280: This is a repetition.
L281-283: Repetition again.
L298-300 and L313-314
Could the author explain the sentence: “Instead of our case where the experimental measurement is not directly related to any of the sensory parameters”.
Do they mean that among the investigated parameters and the NIR data there is not a relation? If it is as reported by the authors, they are correlating two set of data to predict in the future the sensory properties without any chemical (or more generally scientific) soundness. Actually the authors did not comment chemical and physical features of the collect spectra. This step should be included in the discussion and this sentence should be reconsidered.
Could the authors provide details about the sentence: “We have to indicate that at our case the relationship between sensory parameters and NIRS is not so direct”. So there is a relation, but it is not direct. A better explanation should be provided.
Author Response
In the following, we will consider the review 1. We consider that all the comments have been very useful to get a correct version of the paper. Comments for the changes are characterized using “cursive” type
The paper bases on a innovative but difficult task: predict sensory properties, as determined by sensory analysis, by near infrared spectroscopy. The authors presented cheese as case study.
Even though the novelty and originality of the study is relevant, the paper lack of the proper construction. In different cases, as reported in detail below, part of the text should be moved from results to M&M. In any case both sections lack of relevant information as details in the comments below. Moreover, the discussion and results are quite poor in addressing the key point of the research.
The poor English form and the typo made difficult the reading in different paragraphs. Therefore, a major revision of the manuscript is advice.
Introduction
Few consideration about the work cited:
- Instead of ref [3], consider to cite a more recent publication of the same research group. This work consider different animal product analysis (meat, poultry and fish) and led to an ad hoc e-nose system.
Grassi, S., Benedetti, S., Opizzio, M., di Nardo, E., & Buratti, S. (2019). Meat and Fish Freshness Assessment by a Portable and Simplified Electronic Nose System (Mastersense). Sensors, 19(14), 3225.
- I would suggest to substitute the reference [8] with a review in the meat science field, instead of citing a research paper. No interest in one specific, I just suggest few of them, but it is up to the authors the selection.
Chapman, J., Elbourne, A., Truong, V. K., & Cozzolino, D. (2020). Shining light into meat–a review on the recent advances in in vivo and carcass applications of near infrared spectroscopy. International Journal of Food Science & Technology, 55(3), 935-941.
The reference [8] has been replaced by this reference.
Kademi, H. I., Ulusoy, B. H., & Hecer, C. (2019). Applications of miniaturized and portable near infrared spectroscopy (NIRS) for inspection and control of meat and meat products. Food Reviews International, 35(3), 201-220.
The reference to this work has been included at L42
Pasquini, C. (2018). Near infrared spectroscopy: A mature analytical technique with new perspectives–A review. Analytica Chimica Acta, 1026, 8-36.
The reference to this work has been included at L42
Weeranantanaphan, J., Downey, G., Allen, P., & Sun, D. W. (2011). A review of near infrared spectroscopy in muscle food analysis: 2005–2010. Journal of Near Infrared Spectroscopy, 19(2), 61-104.
The reference to this work has been included at L42
Prieto, N., Roehe, R., Lavín, P., Batten, G., & Andrés, S. (2009). Application of near infrared reflectance spectroscopy to predict meat and meat products quality: A review. Meat science, 83(2), 175-186.
The reference to this work has been included at L42
- Please consider to reduce the examples reported for the use of FT-NIR spectroscopy combined with a Back Propagation ANN for the simultaneous analysis of multi-components in food. In detail, consider to reduce the paragraph from L67-L76.
MODIFICATION 3
At lines 67-76 appeared
“The potential of FT-NIR spectroscopy combined with a Back Propagation ANN for the simultaneous analysis of multi-components (carbohydrates, fats and proteins) of pork meat during the bacterial spoiling process was explored in [18]. Also, the synergy of NIRS and neural networks to differentiate Iberian pork from standard Duroc pork in fresh meat was proposed in [19]. The possible applications of ANNs in wine technology are reviewed in [20]. The use of ANNs models to classify apple juice-based commercial beverages was presented in [21]. NIR combined with ANN was used in [22] to predict the quality parameters (protein content, moisture content, Zeleny sedimentation or water absorption) of wheat flour. It is clear that the predicted quality parameters are strongly related to data provided by the NIR. This relationship is not so evident when sensory properties are considered.”
It has been changed to
“The synergy of NIRS and neural networks has been explored in technology of pork meat [18] [19], of flour [20], of wine [21] and other commercial beverages [22].”
- Please consider to reduce the presentation of the aim in the introduction. An attempt is reported below:
L84-90: In this paper, we present a study on the use of ANNs for the estimation of 19 sensory characteristics of samples of controlled-processing cheese. As instrumental measurements we use the NIR spectrum in the range 1100 to 2000 nm. The NIR information was coupled with 19 sensory attributes to predict aspect, taste, smell, texture, and other sensations linked to cheese quality. As a key element of learning process, the training and the prediction ability of the ANN performance is based on all the data provided by a panel of judges trained in the QDA methodology.
MODIFICATION 4
At lines 84-90 appeared
“In this paper, we present a study on the use of ANNs for the estimation of 19 sensory characteristics of samples of controlled-processing cheese (it is currently the most widely consumed dairy product). As instrumental measurements we use the NIR spectrum of the cheese samples, which is based on the absorption of electromagnetic radiation in the band from 1100 to 2000 nm. ANNs will be constructed to predict the numerical value for 19 attributes of the cheese related to the aspect, taste, smell, texture, and other sensations, using NIRS data as inputs. As a key element of learning process, the training and evaluation of the ANN performance is based on all the data provided by a panel of judges trained in the QDA methodology.”
Following the pertinent suggestions of reviewer 1, lines 84-90 have been changed to
“In this paper, we present a study on the use of ANNs for the estimation of 19 sensory characteristics of samples of controlled-processing cheese. As instrumental measurements we use the NIR spectrum in the range 1100 to 2000 nm. The NIR information was coupled with 19 sensory attributes to predict aspect, taste, smell, texture, and other sensations linked to cheese quality. The training and the prediction ability of the ANN performance is based on all the data provided by a panel of judges trained in the QDA methodology.”
Material and Methods
- More details should be provided about the number and weight of cheeses produced. How many cheeses were produced for each formulation? Which was the size of each cheese?
MODIFICATION 5
At lines 106-111 appeared
“Two production processes were carried out: 112 cheeses in the winter season and 112 in the summer with milk collected directly from farms located in Zamora (Spain). Their composition was varied but it was well known: 16 different cheeses were made with percentages between 0% and 100% [26] of raw milk from sheep, goat and cow. After each elaboration process, they were transferred to a drying chamber where the temperature (15_C) and relative humidity (70%) were controlled until the end of its maturation. “
These lines (106-111) have been changed to
“Two production processes were carried out: 112 cheeses in the winter season and 112 in the summer with milk collected directly from farms located in Zamora (Spain).
In a prescribed way, cheeses of 16 different formulations, between 0% and 100%, of raw milk from cow, sheep and goat were made [26]. After a classic initial production process, several pieces of each formulation were taken, initially with a diameter of 10 cm and a thickness of 5 cm. The aging process was performed at a pilot plant with controlled climatic conditions (15ºC and 70% HR).”
- More detail about the amount of sample testes by the panel and by NIR analysis should be provided. How many slices were analysed by NIR, in which point? The NIR signal is highly influence by water distribution and more other changes occurring in the lipid fraction, but this changes are highly influenced by the cheese (sample) geometry, thus further details should be provided. A Figure could better explain the procedure. Also the ANN description is not exhaustive. Figure 2, 3 and 4 seems not to be relevant in the material and methods section, consider to remove them.
At lines 112-115 appeared
“Four and six months after the beginning of each winter and summer drying process, cheese samples were taken from each of the 16 compositions and subjected to both laboratory (NIRS) and sensory analytical testing. Therefore, we have worked with cheeses of 16 different compositions, with milk from two different seasons and different drying times.”
These lines (112-115) have been changed to
“In this way, a period of seven (7) months of aging (from 0.2 to 6 months) was considered to take into account this characteristic in the sensory analysis procedure. Therefore, cheese samples of each of the 16 formulations from winter and summer milks and two aging periods (4 and 6 months) were considered to perform spectral (NIR) and sensory analytical testing, with a total of 64 different samples.”
Further details on NIR and sensory analysis and on the ANN structure will be provided in the corresponding subsections in the results section. However, on lines 129-130 it can be found:
“NIR spectra were recorded by directly applying the fibre optic probe to slices at least 1 cm thick at room temperature (20 - 23 ºC).”
Figures 2, 3 and 4 have been moved from the Materials and Methods section to the Results section. Also the figure 2 and 4 have been resized
Moreover, specific comments are reported in detail below.
- L108: Could be an idea to call the 16 different cheeses, 16 different formulations ”16 different cheeses were made with percentages between 0% and 100% of raw milk from [...].”
MODIFICATION 5
At lines 108 and 113 appeared
“16 different cheeses”
“each of the 16 compositions”
Following the appropriate suggestion, lines 108 and 113 have been changed to
“16 different formulations”
“each of the 16 formulations”
- L110: To what referred the authors with “elaboration process”, production process?
At line 110 appeared
“elaboration process,”
And now
“production process,”
- L113: Consider to substitute “laboratory (NIR)” with “spectral (NIR)”
MODIFICATIONS 6 AND 7
At line 110 appeared
“laboratory (NIR)”
and it has been replaced by
“spectral (NIR)”
- L114-115: Please considered to remove: “Therefore, we have worked with cheeses of 16 different compositions, with milk from two different seasons and different drying times.”
MODIFICATION 8
As suggested above, lines 114 and 115 have been replaced by lines, detailing the number of samples tested by the panel and by the NIR analysis, thus
“Therefore, cheese samples of each of the 16 formulations from winter and summer milks and two aging periods (4 and 6 months) were considered to perform spectral (NIR) and sensory analytical testing, with a total of 64 different samples.”
- L125-127: Here the authors refereed to pictures of the cheeses to evaluate the attributes of appearance. Is it possible to provide more details? Were the pictures produced on the considered formulations? Are the pictures reference materials? In case they were shot by the authors, details should be provided about the place (black box), illuminations and other environmental and resolution conditions.
MODIFICATION 9
The sensory task is described with detail at [27] including the conditions of the performed activities.
[27] González-Martín, I.; Severiano-Pérez, P.; Revilla, I.; Vivar-Quintana, A.; Hernández-Hierro, J.; González-Pérez, C.; Lobos-Ortega, I. Prediction of sensory attributes of cheese by near-infrared spectroscopy. Food Chemistry 2011, pp. 256–263. doi:10.1016/j.foodchem.2010.12.105.
Our work in the present document only makes use of the full data provided by the mentioned work [27]. Therefore, we have modified this section to include only the most significant details needed to understand our work.
At lines 117 and 127 appeared
“The sensory profiling of the cheese was carried out by a panel of 8 tasters trained in the use of the QDA methodology [27], as it provides an objective description of the products in terms of perceived sensory attributes [28]. The panel was composed of 75 % women and 25 % men, aged between 23 and 42. During 18 training sessions, the members were trained (ISO 11035: 1994(E), ISO 4121: 2003(E)) in the sensory profiling of cheese. It was suggested that the group develop a common vocabulary for the evaluation of the sensory characteristics of cheeses. Therefore, the vocabulary generated by the panellists contained simple and specific terms that could be easily memorized and used to discriminate products. The approach described by some authors uses reference scales to assess texture intensity, smell-taste parameters, and it proposes some foods as standards to establish the scale [30]. In accordance with this methodology, photographs were used to evaluate the attributes of appearance that allowed these attributes to be represented at different intensities.”
These lines have (117-127) been changed to
“In [27], the sensory task is described in detail and rigorously, including the conditions of the activities performed. In the present work, we only use the complete data provided by [27]. Therefore, only the most significant details needed to understand the data used in our work will be described below.
A panel of 8 judges was trained in the application of the QDA methodology over 18 sessions. During the training, the panelists agreed on the benchmarks, terminology definitions and evaluation techniques in the sensory profile of the cheese. Sensory properties related to aspect, taste, smell, texture and other sensations were considered. Specifically, there were 19 attributes evaluated (table 1), framed within: aspect (homogeneity, colour, holes), taste (salty, rancid, intensity, sour, buttery), smell (rancid, lactic acid, intensity), texture (chewiness, creaminess, fatty feeling, granularity, hardness), other sensations (hot, remains in throat, retronasal). To quantify the intensity of each attribute, 8-point scale was used, with ’0’ corresponding to the absence of the parameters, ’1’ referring to the minimum intensity and ’7’ to the maximum intensity for each of the parameters. This type of scale was selected because it had given good results in other previous works, when it was used to evaluate the characteristics of the cheeses [30].”
- L130-138: Please provide more details about spectral resolution and/or integration time used to obtain the spectral information. It would be an idea to move here the information reported at line 142 (from 1100 nm to 2000 nm with a separation of 2 nm). Indeed, these information are linked to the NIR analysis and not to the PCA elaboration.
MODIFICATION 10
Line 142 has been moved to this section. In addition, a more descriptive paragraph has been included, which in the previous version of the paper was in the results section
“For each sample, 32 scans were performed, which were averaged to give a spectrum represented as values of log(1/R) (R means reflectance) as a function of the wavelength in the range between 1100-2000 nm with a nominal spectral resolution of 2 nm. To minimize sampling error, all samples were analyzed in triplicate. Prior to each recording, the probe window was cleaned to minimize cross-contamination.”
- L141: Please modify here and in Figure 1 the expression “set of values”, this is a set of variables
Figure 1 has been modified as the reviewer has suggested.
- L140-144: please indicate here the number of samples included in the PCA. An idea could be to write that the data matrix analyzed by PCA was composed by ‘n’ number of samples and ‘m’ variables, that is your 450 variables. Were the data transformed prior to multivariate data analysis? Spectral data, sensory data were pre-treated? Mean centering, Autoscaling which transformation was used? Correction of the spectral signal, i.e. smoothing, baseline correction, MSC, SNV?
In addition to the changes previously commented about this subsection, we have introduced the following paragraph:
“As it has been exposed, 64 samples of cheese were used for this work. At previous section, the wavelength interval used for NIR was presented, so for each sample, 450 variables are considered, that is log(1/R) for each wavelength. The complete set of NIR data for all 64 samples has been taken into account. We consider that, no pre-processing of the spectrum data is necessary to avoid any loss of information. So, the complete and original dataset will be used at this analysis.”
- Information about the software used should be provided here and in the ANN section.
This information has been included at modification 14, as we will comment below.
- Artificial Neural Networks section. This section is not well structured, different details are reported but relevant information are missing or not well described. For instance, from Figure 1 it is clear that PC1, PC2 and PC3 were used, but here no information is provided. No detail about the number of samples used in calibration and prediction is provided and so on. The method description should be highly improved.
MODIFICATION 11
In order to include more relevant information, the following paragraph has been included, along with another paragraph moved from the Results section.
“For each sensory attribute an ANN model will be constructed: in the input layer the PCA data of the cheese sample are entered and in the output layer the estimation of a sensory attribute of the cheese sample is obtained. So, each ANN model will designed using the JavaNNS (Java Neural Network Simulator) application and the model will be developed within the application as a Multi-Layer Perceptron network (MLP).”
MODIFICATION 12
In this sense, it has also been included:
“A training data instance will be a (xi, yi) pair, where xi will be related, using PCA, with the NIR measurement for a cheese sample and yi will be sensory evaluation given for a human tester. So, the training data set will be constituted by 512 pairs (xi, yi): 64 cheese samples with 8 human tasters.”
MODIFICATION 13
The following paragraph has additionally been introduced:
“so that, it will have the capacity for generalization.
A five-fold training procedure will be applied, where a split is made with a selection of 20% instances for validation (randomly selected) and 80% for training, with five (5) training experiments performed.
In this way, the size of the training set, in addition to the network architecture, has influence at its generalization capacity, so it must have a sufficient number of samples and it will be related to the number of neurons in the hidden layer”.
Results
- Measurements of Taster Evaluation Results: Could you describe if the scale was anchored or not.
MODIFICATION 14
To describe in detail the results of the tasting the following paragraph has been introduced in this section
“In particular, the 10-member panel met in 4 tasting sessions to evaluate the cheeses. These 4 sessions correspond to cheeses with 4 and 6 months of maturation and made with winter and summer milk. In each session, cheeses of the 16 formulations were tasted and the 19 attributes described previously were evaluated by the judges. However, punctuations of 2 members were discarded as strongly disagreeing. In total, for each of the attributes, a punctuation data set of 512 items is available”.
- L188: Could you explain the meaning of this sentence: “Then, NIRS will be shown with the following PCA”
MODIFICATION 15
The following paragraph has been added to explain the meaning of this sentence:
“Then, NIR spectra obtained from cheese samples are presented. PCA will be applied to this dataset, i.e. the spectra, to obtain a set of components that will be used for the next stage: the ANN learning task.”
- L205-209. This sentence should be reported in Material and Methods
The sentence L205-209 has been moved to M&M section.
Figure 2 should be moved here, as here it is cited and commented. Indeed, spectral signal should be described and associated to chemical information. Spectral should be coloured according to type of cheese and/or maturation time. There is no need of the colouring solution proposed by the authors.
Figure 3 has been moved here. As indicated, Figure 2 has been moved and spectra have been coloured according the stage and aging time.
- L219: Please move here Figure 3. The sentence “Figure 3 shows these three components over the wavelength.” Is not clear, these are the three PC summarising the information retained in the whole spectral dataset.
Figure 3 has been moved here.
The sentence
“Figure 3 shows these three components over the wavelength”
has been replaced by
“Figure 3 (loadings plot) shows the weights for each the wavelength (variable) when the main components are calculated: the three PC summarising the information retained in the whole spectral dataset.”
- L220-221: They explain most of the variationship at the spectrum: 86 % 12 % and 1 % respectively. What does it means variationship? Variation? Variability? By PCA algorithm the PC1 explains the higher amount of variability and each new component explains less and independent variability (as they are orthogonal). This is by principle, it should not be explain in this way here. Figure 3 and 4 should be moved here and described in detail as loadings plot (Figure 3) and scores plot (figure 4). A good explanation of both graphs is missing. Describe the shape of the spectra, the peaks, the baseline changes for the loadings plot.
Figures 3 and 4 have been moved here.
In the caption of Figure 3 has been included “Loadings plot”
In the caption of Figure 4 has been included “Scores plot”
At lines 218 to 223 appeared
“three (3) score components that can explain the 99.98% of spectral variability are obtained. Figure 3 shows these three components over the wavelength.
They explain most of the variationship at the spectrum: 86 %, 12 % and 1 % respectively. This situation can be also observed graphically at figure 4, where the main variations appear along the first component axis, followed by variations at the second component. Finally, at the third component minimal variations can be observed.”
The paragraph has been rewritten to clarify this aspect
“three (3) score components that can explain the 99.98 % of spectral variability are obtained. They explain most of the variability at the spectrum data: 86 %, 12 % and 1 %, respectively. Loadings of each component are constituted by 450 values, that is, its relationship with the considered 450 variables (each one related with the reflectance at a given wavelength), so loadings of each component would be seen as function of the wavelength.”
Actually PC1 seems to describe a baseline variation; the baseline was corrected somehow? Otherwise the information described is more linked to the roughness of the cheese slices than to real changes in the cheeses. In Figure 4 samples appears quite low in number, how many samples were analysed? For a reliable ANN model you should guarantee an high number of samples in calibration and a worth number in prediction. Samples in the scores plot should be coloured according to a “category”. As mentioned for Figure 2, they could be coloured by type of cheese, maturation time, one or few sensory attribute(s), for example.
MODIFICATION 16
The following paragraph has been included:
“For the training and validation of each of the 19 ANNs, a 512-item punctuation dataset has been available. This set has been split into 80% of the items for the training phase and 20% for the validation phase of each ANN, as explained previously. This number of items in the training dataset and in the validation dataset is sufficient to guarantee a reliable ANN model.”
- L225-227: This sentence should be moved in M&M.
The sentence L225-227 has been moved to M&M section.
- L249-L251: This sentence is relevant for M&M section to describe the data transformation
The sentence L249-L251 has been moved to M&M section.
- L259: Finally here it is reported that the prediction set was composed by 20% of the data. This information should be reported in M&M with the number of data collected, as mentioned in the previous comments.
MODIFICATION 17
L258-260 lines have been eliminated.
- Figure 8. No matter the figure of merits reported (RMSEP and coefficient of determination), it looks that the prediction vs actual values lines (predictors) are quite poor for all the 19 parameters, could the authors explain this?
MODIFICATION 18
The following paragraph has been included
“It can be observed that ANN estimations are mostly inside an acceptable interval (described by red lines defined by the deviation of the variable) around the obtained value given for the set of taster evaluations, pointing out that the expected value is defined by the average per sample of the individual judgements.”
- L274-276 How were acceptable ranges defined?
The answer is included at previous paragraph
- L277-280: This is a repetition.
- L281-283: Repetition again.
Both repeated paragraphs have been removed.
- L298-300 and L313-314
Could the author explain the sentence: “Instead of our case where the experimental measurement is not directly related to any of the sensory parameters”.
MODIFICATION 19
At lines 297 to 300 appeared
“The involved parameters are directly related with the instrumental measures, instead of our case where the experimental measurement is not directly related to any of the sensory parameters. At our case, the parameters related to the aspect obtain acceptable values, homogeneity specially.”
These lines have (297-300) been changed to
“The involved parameters, concerning the perception of fruit colour, are directly related to the instrumental measures, instead of our case where the experimental measurement is not directly related to any of the sensory parameters, because there is no direct property measurement (holes for example).”
Do they mean that among the investigated parameters and the NIR data there is not a relation? If it is as reported by the authors, they are correlating two set of data to predict in the future the sensory properties without any chemical (or more generally scientific) soundness. Actually, the authors did not comment chemical and physical features of the collect spectra. This step should be included in the discussion and this sentence should be reconsidered.
Could the authors provide details about the sentence: “We have to indicate that at our case the relationship between sensory parameters and NIRS is not so direct”. So there is a relation, but it is not direct. A better explanation should be provided.
Previously, these questions have been answered. In fact, Neural Networks have as one of their main advantages the ability to discover indirect (or hidden) relationships, with no need to the finding “direct” or linear dependences that can be known or not. But the relationship exists (although in hidden way) in such a way the predictions work, so ANN can be seen as the tool to discover or use them.

Reviewer 2 Report
General comments:
The present manuscript describes the application of Near Infrared spectroscopy based on Artificial Neural Network to evaluate the sensory attribute of cheese samples. As reported by the authors, no many studies have been published in this field, so this investigation can be considered a promising approach. The same authors reports that additional studies will be necessary to increase the prediction of the results and this manuscript can be accepted as preliminary work. However, several points needs to be improved in both experimental methodology and paper draft to be accepted.
Specific comments:
Generally, the introduction section does not include any table. In this manuscript, the Table 1 can be provided as supplementary material. In fact, this table describes the sensory parameters applied by the taster panel and, in this sense, it can be also inserted in material and methods.
Figure 1: footnotes with the description of abbreviations could be useful for the readers. For example, the authors can provide the extension of PCA and the means of Sensory characteristics and so on. In general, all of the figures could be enriched with detailed descriptions considering the complexity of the study.
Line 185 reports “Results and Discussion” but in the first part of this section only the results are presented; successively, the line 292 reports “Discussion” again. Due to the density of the results, it would be better to keep the two sections well separated to avoid confusion.
In many paragraphs of the results, before presenting the required data, a brief description of the method was indicated which could be moved to the materials and methods. An example could be considered from the line 191 to the line 199.
Figure 8: the readability of the parameters recorded on the axes of each graph would be desirable.
A better explanation of the importance of the direct link between experimental measurement and sensory parameters could be useful (from line 295 to 300).
Please, edit the reference section as indicated by the guidelines for the authors. In particular, the cited journals should be abbreviated.
The reference within the text must be indicated citing the name of the authors. In this manuscript, the authors cite the reference using “in” followed by the number of reference but is not appropriate. An example can be considered from the line 70 to 78.
Author Response
In the following, we will consider the review 2. We consider that all the comments have been very useful to get a correct version of the paper. Comments for the changes are characterized using “cursive” type
General comments:
The present manuscript describes the application of Near Infrared spectroscopy based on Artificial Neural Network to evaluate the sensory attribute of cheese samples. As reported by the authors, no many studies have been published in this field, so this investigation can be considered a promising approach. The same authors reports that additional studies will be necessary to increase the prediction of the results and this manuscript can be accepted as preliminary work. However, several points needs to be improved in both experimental methodology and paper draft to be accepted.
Specific comments:
- Generally, the introduction section does not include any table. In this manuscript, the Table 1 can be provided as supplementary material. In fact, this table describes the sensory parameters applied by the taster panel and, in this sense, it can be also inserted in material and methods.
Table 1 has been removed from the introduction and moved to the "Material and Methods" section.
- Figure 1: footnotes with the description of abbreviations could be useful for the readers. For example, the authors can provide the extension of PCA and the means of Sensory characteristics and so on. In general, all of the figures could be enriched with detailed descriptions considering the complexity of the study.
In figure 1, descriptions of the abbreviations for ease of understanding have been included in the caption.
So, in figure 1, the caption appeared:
“Global scheme of the artificial perception of 19 sensory parameters for each cheese sample”
This text has been changed to:
“Global scheme of the artificial perception of 19 sensory parameters for each cheese sample. PCA1, PCA2 and PCA3 denote the three principal components of Principal Components Analysis (PCA) on NIR spectra. Sensory characteristic $i$ refers to each of the 19 quality parameters of the cheese to be estimated.”
- Line 185 reports “Results and Discussion” but in the first part of this section only the results are presented; successively, the line 292 reports “Discussion” again. Due to the density of the results, it would be better to keep the two sections well separated to avoid confusion.
As suggested by the reviewer, section 3 "Results and Discussion" has been split into section 3 "Results" and section 4 "Discussion”.
- In many paragraphs of the results, before presenting the required data, a brief description of the method was indicated which could be moved to the materials and methods. An example could be considered from the line 191 to the line 199.
“Most of the method descriptions included in the "Results" section have been moved to the "Material and Methods" section, including the lines 191 to 199”
- Figure 8: the readability of the parameters recorded on the axes of each graph would be desirable.
In the axis labels of Figure 8 the font size has been enlarged.
- A better explanation of the importance of the direct link between experimental measurement and sensory parameters could be useful (from line 295 to 300).
At lines 297 to 300 appeared
“The involved parameters are directly related with the instrumental measures, instead of our case where the experimental measurement is not directly related to any of the sensory parameters. At our case, the parameters related to the aspect obtain acceptable values, homogeneity specially.”
These lines have (297-300) been changed to
“The involved parameters, concerning the perception of fruit colour, are directly related to the instrumental measures, instead of our case where the experimental measurement is not directly related to any of the sensory parameters, because there is no direct property measurement (holes for example).”
Please, edit the reference section as indicated by the guidelines for the authors. In particular, the cited journals should be abbreviated.
The reference within the text must be indicated citing the name of the authors. In this manuscript, the authors cite the reference using “in” followed by the number of reference but is not appropriate. An example can be considered from the line 70 to 78.
Reference citation style use the definition provided by SENSORS journal at the style file from MDPI. We have not introduced any modification on the definition, and the final result comes from mpdi.cls file

Round 2
Reviewer 1 Report
Comment 1 - 2.5. Artificial Neural Networks
The implementation of material and methods section lead to a remarkable doubt in the reliability of the obtained results.
Indeed the authors declared to develop the ANN model with the 80% of the total data collected (i.e. 51 samples) and performed and external validation with 13 samples.
Moreover, the authors declared: "In this way, the size of the training set, in addition to the network architecture, has influence at its generalization capacity, so it must have a sufficient number of samples and it will be related to the number of neurons in the hidden layer"
How could they justify that the model built with just 51 samples is stable and reliable, could they also explain how a random selection of 13 samples is not giving a really debatable prediction?
Comment 2 - 3.2. NIRS measurements
Could the authors attempt an association of the spectral bands reported at lines 220-222? Also highlighting which spectral changes are visible during cheese aging.
Authors declared that Figure 2 has been moved and spectra have been coloured according the stage and aging time. No reference in the caption about the identification of aging/color of the spectra.
Moreover, if colors refereed to aging time, the difference visible in the same group of spectra (same color) to what are related? If variety is the answer this part should be deeply commented.
Comment 3 - 3.3 Principal Component Analysis
In general section 3.3 Principal Component Analysis is not thorough.
Are the authors sure that it is relevant to retained the 99.98% of spectral variability? Maybe 1 % explained by the third component is just noise.
Loadings need to be deeply commented and scores should be colored according to aging/cheese or whatever and commented accordingly. The actual visual representation of the scores is meaning less has no consideration can be done.
Comment 4 - 3.5. Evaluating the Performance of Sensory Attribute Predictions
Figure 6 and 7 should use dot for decimal numbering not comma.
Specify if the R2 are refereed to predicion in Figure 7 caption .
Does Figure 8 report prediction or calibration datasets? Maybe both, in this case could the authors provide different color for calibration and validation samples?
Again, it looks that the prediction vs actual values lines (predictors) are quite poor for all the 19 parameters, could the authors explain this? The scatter plots show distribution far to be good prediction, is there something that the reviewer misunderstood?
Furthermore the scales reported in the scatter plots (Figure 8) do not match with the interval of each parameter declared in Table 1 (Table 1. Sensory parameters applied by the taster panel and their main statistical descriptions). For example from table 1 what is reported is :
Property Mean Deviation Minimum Maximum
Aspect-Homogeneity 4.5 1 1.5 7
Aspect-Colour 4.1 1 1.5 7
However in Figure 8 Aspect-Homogeneity ranges between 4.2 and 5.3, the same for Aspect-Colour.
What did occur here?
Comment 5 - Figure 1
Scale and colors of the spectra mismatch. Please remove the scale .
Modify in figure and legend PC1, PC2 and PC3 denote the three principal components of Principal Components Analysis (PCA). They are not PCA but PC (Principal component) as explained.
Author Response
The review made by Reviewer 1 is on the best reviews that the authors remember on their papers. We consider that it helps to improve the readability of the work. In the following, we will consider all the comments provided by Reviewer 1.
Comment 1. 2.5. Artificial Neural Networks
The implementation of material and methods section lead to a remarkable doubt in the reliability of the obtained results.
Indeed the authors declared to develop the ANN model with the 80% of the total data collected (i.e. 51 samples) and performed and external validation with 13 samples.
Moreover, the authors declared: "In this way, the size of the training set, in addition to the network architecture, has influence at its generalization capacity, so it must have a sufficient number of samples and it will be related to the number of neurons in the hidden layer"
How could they justify that the model built with just 51 samples is stable and reliable, could they also explain how a random selection of 13 samples is not giving a really debatable prediction?
This is not an exact information. In fact we have 64 samples that are tasted by 8 taster so, for each parameter we have 512 pair (x,y), with ‘y’ different value (8) for each sample. As anecdote, our first trials in the ANN training considered the value of the mean for the tasters’ panel for each sample (in this case we really have 64 data) and the results were very poor. We have not included this information at the paper, because we consider this choice as faulting one. When we considered the complete dataset (with each individual taster score), the result enhanced a lot. The individual data provided by taster (with the individual noise) played a positive role at the realized experiment, no resampling activity is needed as in other works, nor repeating the samples to adjust the probability distribution of the data.
MODIFICATION 1
So, at LINE 267 in section “Results” we have included:
“64 samples that are tasted by 8 judges. So, for each parameter, we have 512 pair (xi,yi), with (8) different values yi for each sample.”
At LINE 340 in section “Discussion” we have included:
“For the preparation of the ANNs, we have had 19 data sets with 512 items in each: the scores of 19 sensory parameters given by 8 judges for 64 cheese samples. So, we have used 512 pairs (spectral data input, attribute score output) with (8) different values of attribute score output for each sample. In our preliminary experiments for the training of each ANN, we only considered the mean of the tasters' scores for each sample and the results were very poor. Later, when we considered the entire data set (512 items), with the individual scores of each judge, the training of each ANN was possible, and the results improved significantly. The individual data provided by taster (with the individual noise) played a positive role at the realized experiment, no resampling activity is needed, nor repeating the samples to adjust the probability distribution of the data.”
Comment 2 - 3.2. NIRS measurements
Could the authors attempt an association of the spectral bands reported at lines 220-222? Also highlighting which spectral changes are visible during cheese aging.
Previously, our group has made attempts to associate the spectral bands with the maturation of the cheeses. Using only raw information from NIR spectra, without applying any mathematical tools such as ANNs or other, was not possible to find differences among the cheese aging. However, in an earlier paper by the authors using the NIRS and ANN combination, it was possible to successfully classify unknown cheese samples for the aging time, as referred to in the "Introduction" section
“Also, in a work very closely related to that presented in this paper, an ANN was successfully used in [23] to establish the model for determining the drying time and the percentage of milk mixture (cow, goat and sheep) in cheeses of variable composition from data on fatty acids and NIRS.”
[23] Soto-Barajas, M.C.; González-Martín, I.; Salvador-Esteban, J.; Hernández-Hierro, J.; Moreno-Rodilla, V.; Vivar-Quintana, A.M.; Revilla, I.; Lobos-Ortega, I.; Morón-Sancho, R.; Curto-Diego, B. Prediction of the type of milk and degree of ripening in cheeses by means of artificial neural networks with data concerning fatty acids and near infrared spectroscopy. Talanta 2013, pp. 50–55. doi:10.1016/j.talanta.2013.04.043
MODIFICATION 2
It could be interesting to include the aging property as a differentiation in the prediction. We think that it could be quite useful. But at this paper, we want to consider the same properties for the without the differentiation of the aging time. In this work we have tried to process the raw spectrum data to obtain the sensory characteristics. We have included at the section “Discussion” this paragraph:
“It could be interesting to include the aging feature as a differentiation in the prediction. We think that it could be quite useful, and it will be considered at future works.” LINE 362-363
Authors declared that Figure 2 has been moved and spectra have been colored according the stage and aging time. No reference in the caption about the identification of aging/color of the spectra.
MODIFICATION 3
We have included the color for the aging. As suggested by Reviewer, we have proceeded to include in the caption of Figure 2, the identification aging/color of the spectra.
“Four colors are used to distinguish the spectra according to the aging and seasons of the cheese (4 months/Winter-Green, 6 months/Winter-Magenta, 4 months/Summer-Yellow, 6 months/Summer-Cyan”
Moreover, if colors refereed to aging time, the difference visible in the same group of spectra (same color) to what are related? If variety is the answer this part should be deeply commented.
As indicated by the Reviewer, the variety is the visible difference in the same group of spectra (same color). It is possible to obtain an estimate of the variety from the spectra. In this sense, the authors have used different mathematical tools for its estimation. In [23], there is the work published for the determination of the formulation of cheeses from NIRS by means of ANNs. However, in this study, the composition of fatty acids was also tested as an instrumental measure. As a result, it was found that NNA-based predictive models discover a greater relationship between formulation and fatty acid composition than between formulation and NIRS data.
In fact, NIR information is larger than the information that we try to extract. This work tries to find a “small” amount of the hidden information include at the NIR spectra using MLP ANN.
Comment 3 - 3.3 Principal Component Analysis
In general section 3.3 Principal Component Analysis is not thorough.
Are the authors sure that it is relevant to retained the 99.98% of spectral variability? Maybe 1 % explained by the third component is just noise.
MODIFICATION 4
In fact, the third component has less weight that the first and second one, but at a non-linear behavior we do not know if this “small” weight incorporates useful information. In future works we will consider the comparation for a performance for ANN with two or three inputs
At Discussion section (line 334) we have included this paragraph:
“The spectral data were incorporated into the ANNs by means of Principal Component Analysis. With the first three components, 99.98% of the spectral variability can be explained: 86%, 12% and 1%, respectively. From a comparative point of view, the weight of the third component PC3 may be negligible compared to the weight of PC1 and PC2. However, we have also included the third component as an input in the ANN, since we think that in a non-linear behavior this “small” weight can be significant, and the third component can incorporate useful information.”
Loadings need to be deeply commented and scores should be colored according to aging/cheese or whatever and commented accordingly. The actual visual representation of the scores is meaning less has no consideration can be done.
Comment 4 - 3.5. Evaluating the Performance of Sensory Attribute Predictions
MODIFICATION 5
Figure 6 and 7 should use dot for decimal numbering not comma.
Figures 6, 7 and 9 have been modified: point for decimal numbering is now used
Specify if the R2 are refereed to prediction in Figure 7 caption.
The caption in figure 7 has been changed in this way at the caption
”Squared correlation coefficient of prediction for the whole set of parameters for unknown samples”
Does Figure 8 report prediction or calibration datasets? Maybe both, in this case could the authors provide different color for calibration and validation samples?
In figure 8 the prediction dataset has been considered. We have also modified the caption of the figure to explain that it refers to the prediction data set:
“Scatter plot for PCA-ANN prediction results vs tasters' scores for all sensory attributes for the prediction dataset. The performance over the attribute span is observed for the whole set of attributes in the prediction stage. The green line is the expected score and the red lines mark the interval defined with the deviation of the tasters' scores for the total set of samples”
Again, it looks that the prediction vs actual values lines (predictors) are quite poor for all the 19 parameters, could the authors explain this? The scatter plots show distribution far to be good prediction, is there something that the reviewer misunderstood?
MODIFICATION 6
As we had already included at lines 289-294, most of the values of the predictions are inside the interval defined by the deviation in judges' scores (Table 1) for each attribute. So, ANN prediction behaviour is similar (or better) than taster. This is the interpretation that we made of the fact that the predictions are inside the region defined by the deviation (red lines around the green line) in judges' scores (Table 1).
However, we have improved that paragraph, as follows (LINE 287):
“To illustrate the predictive capacity of each ANN, the predicted scores provided for each parameter are presented (figure 8 - scatter plot) on the same graph with the judges' scores for the prediction dataset. It can be observed that ANN predictions are mostly inside an acceptable interval around the expected value for the set of taster evaluations, pointing out that the expected value is defined by the mean per sample of the individual judgements. This interval is marked within each figure by red color lines, defined by the deviation in judges' scores for each attribute, with the data collected in the table 1. So, ANN prediction behaviour is similar than taster.”
Furthermore the scales reported in the scatter plots (Figure 8) do not match with the interval of each parameter declared in Table 1 (Table 1. Sensory parameters applied by the taster panel and their main statistical descriptions). For example from table 1 what is reported is:
Property Mean Deviation Minimum Maximum
Aspect-Homogeneity 4.5 1 1.5 7
Aspect-Color 4.1 1 1.5 7
However in Figure 8 Aspect-Homogeneity ranges between 4.2 and 5.3, the same for Aspect-Color.
What did occur here?
MODIFICATION 7
In order to show the performance of ANN at a plot, we have considered, as judges' score in a given sample, the mean of the judges' individual scores in each sample and not the individual judges' score (this is the information described at Table 1). In the paper this fact was pointed out when defining “scatter plot” at the section “2.6. Metrics to Evaluate the ANN Model” where it appeared
“Scatter plot of predicted values and judges’ score mean for each sample.”
This is the reason why in Figure 8 the value range for the real scores does not coincide with the data collected in Table 1. The X axis of the scatter plot does not show the extreme values "1" and "7" of the individual judges' scores for each sample, as we have considered the average judges' scores per sample.
So at LINE 288 has been changed to
“on the same graph with the mean judges' scores for the prediction dataset”.
The same type of graph was used in an earlier work by the authors when trying to predict sensory attributes using modified partial least squares (MPLS) regression, although the results were worse, as reported in the section “Discussion”
“It can be observed that our results, obtained with ANNs on unknown samples, are better than those obtained in [25].”
[25] González-Martín, I.; Severiano-Pérez, P.; Revilla, I.; Vivar-Quintana, A.; Hernández-Hierro, J.; González-Pérez, C.; Lobos-Ortega, I. Prediction of sensory attributes of cheese by near-infrared spectroscopy. Food Chemistry
In addition, we have solved an error when changing the size of the labels and ticks on the axes of the figures, as also was suggested by Reviewer 2. This error in Figure 8 has been corrected
Comment 5 - Figure 1
MODIFICATION 8
Scale and colors of the spectra mismatch. Please remove the scale.
Figure 1 has been modified: the scale removed
Modify in figure and legend PC1, PC2 and PC3 denote the three principal components of Principal Components Analysis (PCA). They are not PCA but PC (Principal component) as explained.
Figure 1 and the caption have been modified